# Review and Mechanism of the Thickness Effect of Solid Dielectrics

**DOI:** 10.3390/nano10122473

**Published:** 2020-12-10

**Authors:** Liang Zhao, Chun Liang Liu

**Affiliations:** 1Science and Technology on High Power Microwave Laboratory, Northwest Institute of Nuclear Technology, P.O. Box 69 Branch 13, Xi’an 710024, China; zhaoliang@stu.xjtu.edu.cn; 2Key Laboratory of Physical Electronics and Devices of Ministry of Education, Xi’an Jiaotong University, No. 28 West Xianning Rd., Xi’an 710049, China

**Keywords:** thickness effect, solid insulation dielectrics, breakdown strength

## Abstract

The thickness effect of solid dielectrics means the relation between the electric breakdown strength (*E_BD_*) and the dielectric thickness (*d*). By reviewing different types of expressions of *E_BD_* on *d*, it is found that the minus power relation (*E_BD_* = *E*_1_*d*
^−*a*^) is supported by plenty of experimental results. The physical mechanism responsible for the minus power relation of the thickness effect is reviewed and improved. In addition, it is found that the physical meaning of the power exponent *a* is approximately the relative standard error of the *E_BD_* distributions in perspective of the Weibull distribution. In the end, the factors influencing the power exponent *a* are discussed.

## 1. Introduction

Solid dielectrics are widely used in high-voltage (HV) devices and pulsed power systems [1,2,3,4,5,6]. The breakdown characteristics of solid dielectrics have been researched for nearly 100 years. The classical solid dielectric breakdown theory includes intrinsic breakdown, avalanche breakdown, thermal breakdown, and electro-mechanical breakdown, which were established by A. von. Hippel [7,8,9,10,11,12,13,14], H. Frohlich [15,16,17,18,19,20,21], and F. Seitz [22,23,24,25,26,27,28] and later refined by G. C. Garton [29], S. Whitehead [30,31], and J. J. O’Dwyer [32,33,34]. The classical breakdown theory mainly focuses on questions such as the difference between intrinsic breakdown and avalanche breakdown, the relation between electric breakdown strength (*E_BD_*) and dielectric thickness (*d*), the dependence of *E_BD_* on temperature (*T*), and the tendency of *E_BD_* in applied waveforms, etc. Table 1 summarizes these types of breakdown.

From this table, it can be seen that a common question is the dependence of *E_BD_* on dielectric thickness *d*. If *E_BD_* is independent of *d*, the breakdown may be classified as the intrinsic type; if not, the breakdown may be classified to other types. Thus, research on the thickness effect of solid dielectrics is important in breakdown theory. In practice, research on the thickness effect is also important for practical insulation design, since *E_BD_* is directly related to the size (namely, thickness) and the lifetime of solid insulation structures.

In view of these considerations, understanding the mechanism of the thickness effect and knowing the specific expression of *E_BD_* on *d* are meaningful. However, research on these topics is not ideal and systematic, even thought a lot of experiments have been reported.

The main questions can be classified into the following aspects:

(1) The relation between *E_BD_* on *d* is not unified. For example, at least four expressions related to the effect of *E_BD_* on *d* have been reported.

(2) The mechanism responsible for the thickness effect is not well understood. This is probably why different expressions exist.

(3) The application condition for some types of *E_BD_*-*d* expressions is not clear.

Keeping in mind these questions above, a review and theoretical analysis are presented in this paper. In Section 2 and Section 3, different expressions regarding the relation of *E_BD_* with *d* are reviewed and compared, aiming to find the most appropriate one. In Section 4, the physical model for the minus power relation of the thickness effect, which is considered as the most appropriate one, is reviewed and improved. In Section 5, the thickness effect is interpreted in light of the statistics distribution. In Section 6, the power exponent in the minus power relation of the thickness effect is discussed. The last question is discussed in the conclusions of this paper.

Before starting the review, it is noted that “solid dielectrics” in this paper mainly denote the solid insulation dielectrics, which cannot be easily transformed and can play both the role of insulation and support in HV devices, such as polymethyl methacrylate (PMMA), polyethylene (PE), nylon, Al_2_O_3_, MgO, TiO_2_. Solid dielectrics of elastomers such as polymerized styrene butadiene rubber, cis-polybutadiene, and polyisoprene rubber are not discussed in this paper [35]. This is not only because these types of dielectrics have an unstable thickness but also because there are several breakdown mechanisms involved in the failure process aside from electric breakdown [36], such as thermal breakdown and electro-mechanical breakdown.

## 2. Review on Reported Relations about *E_BD_* on *d*

At least four types of relations between *E_BD_* and *d* have been reported in the literature—i.e., constant relation, reciprocal-single-logarithm relation, minus-single-logarithm relation, and double logarithm relation. Each relation has its own experimental supports.

### 2.1. Constant Relation

The constant relation means that *E_BD_* is equal to a constant *C*, which is independent on *d*—i.e.,
(1)$$E_{BD}\left(d\right)=C.$$

Two groups of experimental results support this type of relation. The first group comes from W. J. Oakes [6] in 1948, who tested the breakdown voltage *U_BD_* of PE samples ranging from 0 to 0.2 mm under dc conditions. He found that *U_BD_* was linearly dependent on *d*, which means that *E_BD_* is a constant. The second group of experimental results comes from J. Vermeer [37] in 1954, who tested the *E_BD_* of Pyrex glass and found that, when the temperature is −50 °C or the voltage rise time *t_r_* is equal to 10^−5^ s, *E_BD_* would be independent of *d*. These two groups of experimental results are re-analyzed and re-plotted in Figure 1.

The constant relation of *E_BD_* on *d* may reflect some intrinsic breakdown characteristics of dielectrics.

### 2.2. Reciprocal-Single-Logarithm Relation

The reciprocal-single-logarithm relation means that 1/*E_BD_* is linear to lg*d*—i.e.,
(2)$$1/E_{BD}\left(d\right)=A\mathrm{lg}d-B.$$
where *A* and *B* are constants. This relation is based on the “40-generation-electron theory”, which was put forward by F. Seitz in 1949 [38]. The deduction process for Equation (2) is presented concisely as follows: a seed electron can prime an electron avalanche after 40-time impact and ionization—i.e.,
(3)αd=40,
where α is the ionization coefficient, which can be written as follows:(4)$$\alpha \propto \exp\left(-\frac{E_{H}}{E}\right),$$
where *E_H_* is a parameter with the same unit of electric field and is dependent on *E*. Inserting Equation (4) into Equation (3) gives the following:(5)$$d\exp\left(-\frac{E_{H}}{E}\right)=d_{0}\left(E\right),$$
where *d*_0_(*E*) means a unit thickness dependent on the applied field *E*. Based on Equation (5), *E* is solved and defined as *E_BD_*, which is as follows:(6)$$E_{BD}=\frac{E_{H}}{\ln\left[d/d_{0}\left(E_{BD}\right)\right]}.$$

As to Equation (6), both sides have *E_BD_* and this equation cannot be solved. Even still, it is widely accepted that *E_H_* and *d*_0_(*E*) can be considered as constants. Based on this, Equation (6) can be further changed to:(7)$$\frac{1}{E_{BD}}=\frac{1}{E_{H}}\ln d-\frac{\ln d_{0}}{E_{H}}.$$

By comparing (7) and (2), one can see that *A* = 1/*E_H_* and *B* = 1/*E_H_*ln*d*_0_.

Additionally, there are two groups of experimental results supporting this type of relation. The first group comes from A. W. E. Austen in 1940 [30], who tested the *E_BD_* of clean ruby muscovite mica dependent on a thickness in a range of 200–600 nm under dc conditions. He found that:(8)$$E_{BD}=\frac{54}{\ln\left(d/d_{0}\right)}\left(\mathrm{MV}/\mathrm{cm}\right),$$
where *d*_0_ = 5 nm. The second group comes from J. J. O’Dwyer in 1967 [33], who thoroughly analyzed the electron-impact-ionization model of Seitz and summarized the relevant experimental results in history to support the *E_BD_*-*d* relation in (2). Figure 2 replots the two groups of experimental results with 1/*E_BD_* dependent on lg*d*.

### 2.3. Minus-Single-Logarithm Relation

The minus-single-logarithm relation means that *E_BD_* is linear to lg*d* with a minus slope rate—i.e.,
(9)$$E_{BD}\left(d\right)=D-F\mathrm{lg}d,$$
where *D* and *F* are constants. In 1963, R. Cooper tested the *E_BD_* of PE with a thickness ranging from 25 to 460 μm on a millisecond time scale [39]. After fitting, Cooper gave the following relation:(10)$$E_{BD}\left(d\right)=12.8-3\mathrm{lg}d\left(\mathrm{MV}/\mathrm{in}\right),$$
where *d* is in inches. It is noted that this result was published in *Nature* but there was no theoretical basis for Equation (9) or Equation (10). Figure 3 replots the experimental results with the units of MV/cm and cm.

### 2.4. Double-Logarithm Relation

The double-logarithm relation means that lg*E_BD_* is linear to lg*d*—i.e.,
(11)$$\mathrm{lg}E_{BD}\left(d\right)=G-a\mathrm{lg}d.$$
where *G* and *a* are constants. The expression in Equation (10) is also called the minus power relation since it can be transformed into:(12)$$E_{BD}\left(d\right)=E_{1}d^{-a},$$
where *E*_1_ = 10^G^. In 1964, F. Forlani put forward a model for the thickness effect by taking into account the electron injection from cathode and the electron avalanche in dielectrics together [40,41]. After a series deduction, he found that:(13)$$E_{BD}\approx \frac{C_{\phi }}{d^{a}}.$$

According to Forlani, *C_φ_* and *a* are constants and *a* ranges from 1/4 to 1/2. By re-analyzing the experimental data from R. C. Merrill on Al_2_O_3_, he verified this model. Here, we also replot this group of data in a log-log coordinate system, as shown in Figure 4.

As a summary of this section, Table 2 lists the expressions, mechanisms, and researchers for the formulae of the thickness effect in the literature.

## 3. Comparison of Different *E_BD_-d* Relations

Now a question comes: which one is more appropriate to characterize the thickness effect of solid dielectrics? In this section, these four types of *E_BD_*-*d* relations are compared in order to find the most appropriate one.

### 3.1. More Results on lgE_BD_-lgd Relation

When reviewing the thickness effect of solid dielectrics, it is found that plenty of experimental results have been reported on this topic. Among these groups of data, some were already fitted with the lg*E_B_*_D_-lg*d* relation and were given the power exponent, such as those reported by J. H. Mason [43,44,45], G. Yilmaz [46,47], and A. Singh [48,49]; other groups of data were re-analyzed in order to obtain the minus power exponent, such as the data reported by Y. Yang [50], K. Yoshino [51], K. Theodosiou [52], and G. Chen [53]. As a summary, Table 3 lists the researcher, test object, thickness range, value of *a*, and feature of each group of experiments. Based on this table, the distribution of *a* in a wide range of thickness is plotted, as shown in Figure 5.

With Table 3 and Figure 5 together, some basic conclusions can be drawn:

(1) The minus power relation for the thickness effect of solid dielectrics holds true in a wide thickness range from several Å s to several millimeters, even though the power exponent *a* is different.

(2) The value of *a* in the minus power relation ranges from 0 to 1. The largest value is about 1, see the lines of 1971/Agarwal [54] and 2003/Yang [50]. The smallest value is 0.022; see the line of 2012/Chen. If other factors are neglected, *a* is averaged at about 0.5.

It is worth mentioning that the distribution of *a* in thickness seems “random”. This will be discussed especially in Section 7.

According to Section 3.1, it can be seen that the number of experimental groups needed to support the minus power relation is far more than those needed to support three other types of thickness–effect relations. Thus, it is necessary to make a comparison between the minus power relation and other three types of relations on the thickness effect.

### 3.2. Comparison between Minus Power Relation and Other Three Types of Relations

#### 3.2.1. Comparison with the Reciprocal-Single-Logarithm Relation

Figure 6 directly shows the fitting results on the raw experimental data from Austen [30] with the double-logarithm relation and the reciprocal-single-logarithm relation. From Figure 6a, it seems that the two types of fitting have no difference due to the distribution and error bar of the raw data. In order to show the fitting results clearly, the error bar is removed and only the average data are plotted in a log-log coordinate system, as shown in Figure 6b. From this figure, it is seen that the reciprocal-single-logarithm relation gives a smaller *E_BD_* in the middle thickness range, whereas it presents a larger *E_BD_* in the lower and the higher thickness ranges.

Aside from the experimental data from Austen, the data from O’Dwyer are also compared, as shown in Figure 7a,b. From these two figures, it can be seen that the minus power relation gives a better fit in all the data ranges, whereas the reciprocal-single-logarithm relation can only cover parts of the experimental data range. In addition, there is a pre-condition to use the reciprocal-single-logarithm relation. Because log*d*/*d*_0_ should be positive, *d* should be larger than *d*_0_, or else a minus *E_BD_* would be result from this.

Due to these considerations, it is believed that the minus power relation is preferable to the reciprocal-single-logarithm relation to describe the thickness effect of solid dielectrics.

#### 3.2.2. Comparison with the Minus-Single-Logarithm Relation

Similarly, the raw experimental data from R. Cooper are used and fitted with the minus power relation and the minus-single-logarithm relation, respectively, which are shown in Figure 8. From this figure, it can be seen that both the two types of relations can pass the main data range. However, the *E_BD_* fitted in the lower and the higher thickness ranges by the minus-single-logarithm relation is lower than that fitted by the minus power relation, and the deviation becomes greater as *d* deviates from the average thickness of 100 μm; whereas, the minus power relation can be applied in a wide thickness range. In view of this, the minus power relation is also believed to be preferable to the minus-single-logarithm relation. It is noted that the dispersion of data is very high. This is probably due to the two types of waveforms applied (1/8000 μs and 1/120 μs), the different grades of polythene (sample material), the faults in the sample, and the systematic errors in measurement [39].

#### 3.2.3. Comparison with the Constant Relation

It is believed that the constant relation for the thickness effect is just an extreme case of the minus power relation when *a* is close to 0. As to Equation (13), if *a* = 0, *E_BD_*(d) = *C_φ_*, which is a constant.

This transition from the minus power relation to the constant relation can be verified by the experimental results by G. Chen in Table 3 [53], since the value of *a* is 0.022, which means a much weaker thickness effect.

As a conclusion for this section, the minus power relation is believed to be preferable to the other three types of relations to describe the thickness effect on *E_BD_*.

## 4. Mechanism for the Minus Power Relation

If the minus power relation is the most appropriate relation to describe the thickness effect, what is the potential mechanism? As mentioned previously, F. Forlani put forward a physical model to explain the thickness effect and deduced the minus power relation with a theoretical power exponent from 1/4 to 1/2 in 1964. However, the practical range of *a* is from 0 to 1. This deviation needs to be analyzed and discussed. Before the analysis, the physical model by Forlani is reviewed first.

### 4.1. Review on Model Suggested by F. Forlani

In Forlani’s model, the electrodes and the dielectric were considered together for the occurrence of a breakdown [40,41]. The basic starting point of this model can be written as follows:(14)$$j\left(d\right)=j_{i}P\exp\left(\alpha d\right),$$
where *j*(*d*) represents the current density when the seed electrons leave the cathode with a distance of *d*; *j_i_* represents the current density near the cathode; *P* denotes the probability for electrons to change from the stable state to an unstable state, which can also be considered as the probability of an avalanche forming; *α* is the ionization coefficient, which means that *α* times of impacts to the atoms can take place when an electron moves a distance of 1 cm along the inverse field direction in dielectrics; exp(*αd*) represents the increasing times for one seed electron moving along a distance of *d*. The physical meaning of Equation (14) can be explained as follows: *j_i_* electrons of an initial electron number of *j*_0_ are emitted from the cathode to a dielectric due to the weakness of the potential barrier of the cathode; after moving a distance of *d* in the dielectric, the electron number becomes *j_i_*exp(*αd*) due to the impact ionization and multiplication of exp(*αd*). The final electron number in the avalanche head is *j_i_*exp(*αd*)*P* due to the avalanche formation probability *P*. The schematics of this model are expressed in Figure 9. Namely, there are basically two steps for a breakdown to take place: firstly, the electron injection process of cathode, which is written as Step 1; secondly, the avalanche process in dielectrics, which is written as Step 2. The second step combines two sub-processes together—the electron multiplication process and the avalanche formation process—which are written as Step 2a and Step 2b. It is noted that Step 2a and Step 2b in practice cannot be divided. Here, this is just for the convenience of the deduction of *E_BD_* in Section 4.2.

If the current density (or electron number) increases to a critical level *j_BD_*, which can evaporate or erode the local dielectric, Forlani believed that breakdown occurs—that is:(15)$$E_{BD}=E{|}_{j\left(d\right)=j_{BD}}.$$

Taking into account Equation (15) for Equation (14) and making a logarithm transformation for Equation (14) gives:(16)$$\ln j_{i}+\alpha d+\ln P=\ln j_{BD}.$$

Now, each part in the left side of Equation (16) is specially analyzed.

(1) As to *j_i_*, it represents Step1 and is related to the way of electron injection. Forlani takes into account two typical means of electron injection.

The first way is field-induced emission. When the applied field is higher than 1 MV/cm, the field-induced emission is mainly the tunnel effect, which can be expressed as follows:(17)$$j_{i}=AE^{2}\exp\left(-\frac{B}{E}\theta \left(y\right)\right),$$
or
(18)$$\ln j_{i}=\ln\left(AE^{2}\right)-\frac{B\theta \left(y\right)}{E},$$
where *θ*(*y*) is a modified factor which is related to temperature *T*. If the *T* is fixed, *θ*(*y*) can be considered to be a constant.

The second way is the field-assisted thermal emission. Based on the Schottky effect, *j_i_* can be expressed as follows:(19)$$j_{i}=j_{0}\exp\left(-\frac{\Phi }{kT}+\frac{0.44}{T}E^{1/2}\right),$$
or
(20)$$\ln j_{i}=\ln j_{0}-\frac{\Phi }{k_{B}T}+\frac{0.44}{T}E^{1/2},$$
where *Φ* is the potential barrier of the cathode and *k_B_* is Boltzmann’s constant.

(2) As to *αd*, it represents Step 2a and Forlani believed that *α* was related to the applied field *E*; this can be written as follows:(21)$$\alpha d=\frac{eEd}{\Delta I},$$
where *e* is the absolute electron charge and Δ*I* is the ionization energy.

(3) As to avalanche formation probability *P*, it represents Step 2b. Forlani solved the wave number equation:(22)$$\frac{1}{k}\bar{{\left(\frac{dk_{x}}{dt}\right)}_{sc}}\bar{{\left(\frac{d\Delta I}{dt}\right)}_{sc}}={\left(\frac{eE}{m*}\right)}^{2},$$
where *k* is the wave vector, *k_x_* is the wave vector at a distance of *x*, and *m** is the effective mass of the electron. He obtained the curve of *P* as a function of *E*, which is replotted in Figure 10. In this figure, *E_H_* represents the breakdown field deduced from the Frohlich low-energy criterion.

### 4.2. Solution and Improvements for Forlani’ Model

Forlani solved Equation (16) in three cases. First, when the dielectric thickness *d* was small and the electron injection was mainly via the tunnel effect. According to Forlani, *d* should be far smaller than the electron recombination length *x*_0_. In this case, Equation (18) can be inserted into Equation (16), i.e.,
(23)$$\frac{eEd}{\Delta I}-\frac{B\theta \left(y\right)}{E}+\ln\left(AE^{2}\right)+\ln P=\ln j_{BD}.$$

Since the applied field is large, *P* is close to 1 and ln*P* = 0. In addition, ln(*AE*^2^) and ln*j_BD_* are smaller parts compared with the first two parts in the left side of Equation (23). Thus, they can be ignored. Taking into account all of these, Equation (23) can be simplified to be:(24)$$\frac{eEd}{\Delta I}-\frac{B\theta \left(y\right)}{E}\approx 0.$$

Thus, the solution for Equation (24) is:(25)$$\begin{array}{ccc} E_{BD}\approx E_{TN}d^{-\frac{1}{2}} & \mathrm{where} & E_{TN} \end{array}={\left(\frac{B\theta \left(y\right)}{e}\right)}^{\frac{1}{2}}.$$

In Equation (25), *E_TN_* is a constant which is related to the tunnel effect.

Second, when the dielectric thickness is small and the electron injection is mainly via the Schottky effect, Equation (20) can be inserted into Equation (16), which gives:(26)$$\frac{eEd}{\Delta I}-\frac{\Phi }{kT}+\frac{0.44}{T}E^{1/2}+\ln P+\ln j_{0}=\ln j_{BD}.$$

Similarly, since *E* is large, ln*P* is close to 0. In addition, Forlani believed that, by reasonable estimation, only the first two parts in the left side of Equation (25) are predominant—i.e.,
(27)$$\frac{eEd}{\Delta I}-\frac{\Phi }{kT}\approx 0.$$

Solving (27) gives:(28)$$\begin{array}{ccc} E_{BD}\approx E_{ST}d^{-1} & \mathrm{where} & E_{ST} \end{array}=\frac{\Phi \Delta I}{k_{B}Te},$$
where *E_ST_* is a constant which is related to the Schottky effect.

Third, when the dielectric thickness is large and the electron injection from cathode can be neglected, only the electron multiplication part *αd* and the avalanche formation probability ln*P* in Equation (16) need to be taken into account. Additionally, by neglecting ln*j_BD_*, one can obtain:(29)$$\frac{eEd}{\Delta I}+\ln P\approx 0.$$

In order to solve Equation (29), the specific effect of ln*P* on *E* should be known in advance. Unfortunately, Forlani did not give the theoretical relation of *P* with *E*, only presenting the curve of (−ln*P*) and *E*, as shown in Figure 10. In addition, Forlani believed that −ln*P* is proportional to 1/*E*^3^. Those are all the clues about *P* and *E* supplied by Forlani.

By analyzing these clues, it is found that lg(−ln*P*) should be linear to lg*E* with a slope rate of −3. Thus, we get the raw data in Figure 10 and replot the data of (−ln*P*, *E*) in a log-log coordinate system, as shown in Figure 11. From this figure, it can be seen that lg(−ln*P*) is really linear to lg*E* with a slope rate of −3, but this relation only holds true within *E/E_H_* < 0.8. In order to present an accurate fit, we follow the method suggested by Forlani and consider each segment of the curve in Figure 11 to be a line and fit these segmental lines with a different minus power relation.

Here, a five-segment approximate curve is given, which is:(30)$$-\ln P=\{\begin{array}{c} \begin{array}{cc} \frac{K_{1}}{E^{3.0}} & \left(0.45E_{H}\leq E\leq 0.8E_{H}\right) \end{array}\\ \begin{array}{cc} \frac{K_{2}}{E^{5.1}} & \left(0.80E_{H}<E\leq 0.9E_{H}\right) \end{array}\\ \begin{array}{cc} \frac{K_{3}}{E^{8.7}} & \left(0.9E_{H}<E\leq 0.95E_{H}\right) \end{array}\\ \begin{array}{c} \begin{array}{cc} \frac{K_{4}}{E^{26}} & \left(0.95E_{H}<E\leq 0.99E_{H}\right) \end{array}\\ \begin{array}{cc} \frac{K_{5}}{E^{213}} & \left(0.99E_{H}<E\leq 1.0E_{H}\right) \end{array} \end{array} \end{array}.$$

As to Equation (30), two points need to be clarified: (1) the more segments are set, the more accurate the approximate curve is; (2) the closer *E* is to *E_H_*, the larger the power exponent is.

As to each segment of the fitting curve, the following relation can be obtained:(31)$$\begin{array}{cc} -\ln P=\frac{K_{i}}{E^{\omega_{i}}} & p_{i}<E/E_{H}\leq \end{array}p_{i+1}.$$

In this range of *E,* we solve Equation (29) and obtain:(32)$$\begin{array}{ccc} E_{BD}\approx E_{i}d^{-\frac{1}{\omega_{i}+1}} & \mathrm{where} & E_{i} \end{array}={\left(\frac{\Delta IK_{i}}{e}\right)}^{\frac{1}{\omega_{i}+1}}.$$

If *m* is equal to *ω_i_* + 1, Equation (32) changes to:(33)$$\begin{array}{ccc} E_{BD}\approx E_{i}d^{-\frac{1}{m}} & \mathrm{where} & m=\omega_{i}+1 \end{array}.$$

Equation (33) means that the relation between *E_BD_* and *d* conforms to a minus power relation with a power exponent of 1/*m* as the dielectric thickness increases. In addition, the smallest *m* is 4 in Equation (33) or the largest *a* is 1/4 for Equation (12) based on Figure 12.

By generalizing the three cases of solutions in Equation (25), Equation (28) and Equation (33) together, it can be concluded that the thickness effect of solid dielectric meets the minus power relation—that is:(34)$$E_{BD}=E_{k}d^{-1/m}.$$
where *E_k_* and *m* are both constants. When *m* = 1 or *a* = 1, this relation represents a breakdown mechanism related to the Schottky effect; when *m* = 2 or *a* = 1/2, this relation represents a breakdown mechanism related to the tunnel effect; when *m* ≥ 4 or *a* ≤ 1/4, the breakdown mechanism is related to the avalanche process. Figure 12 shows the three types of breakdown mechanism for the minus power relation with *m* as the argument.

## 5. Minus Power Relation from Weibull Statistics

The above analysis is just from the perspective of the breakdown mechanism. It is also necessary to analyze the thickness effect from the perspective of statistical distribution.

### 5.1. Deduction for the Minus Power Relation

The Weibull distribution is a widely-used method for analyzing failure events [63], especially for the breakdown in insulation dielectrics [64,65,66,67,68]. The two-parameter Weibull distribution is as follows:(35)$$F\left(E\right)=1-\exp\left(-\frac{E^{m}}{\eta }\right),$$
where *F*(*E*) is the breakdown probability, *E* is the applied field; *m* and *η* are the shape parameter and the dimension parameter, respectively. If *E* is equal to *η*^1/*m*^, *F*(*E*) = 0.6321. Moreover, this field is defined as *E_BD_*. Now, assume that the reliability of a solid dielectric with a thickness of *d*_1_ is *R* and this sample is placed in a field of *E*. Since *F* + *R* = 1, *R* can be expressed as follows:(36)$$R\left(E\right)=\exp\left(-\frac{E^{m}}{\eta }\right).$$

Further, assume that *M* samples with the same thickness and configuration are placed in series in the field of *E* [69].

By neglecting the electrode effect and edge effect and by assuming that the reliability of each sample is equal, the following relation would hold true:(37)$$R_{M}\left(E\right)=\exp{\left(-\frac{E^{m}}{\eta }\right)}^{M}=\exp\left(-\frac{E^{m}}{\eta /M}\right).$$

Then, the breakdown probability of the thick sample is:(38)$$F_{M}\left(E\right)=1-R_{M}\left(E\right)=1-\exp\left(-\frac{E^{m}}{\eta /M}\right).$$

Based on the definition of *E_BD_* from the perspective of the Weibull distribution, the breakdown strength *E_BDM_* for the thick sample is:(39)$$E_{BD_{M}}={\left(\frac{\eta }{M}\right)}^{\frac{1}{m}}={\left(\frac{1}{M}\right)}^{\frac{1}{m}}\cdot \eta^{{}^{\frac{1}{m}}}.$$

Taking into account the assumptions that *d_M_* = *Md*_1_ and *η*^1/*m*^ = *E_BD_*_1_, Equation (39) is simplified as:(40)$$E_{BD_{M}}=\frac{d_{1}^{\frac{1}{m}}E_{BD1}}{d_{M}^{\frac{1}{m}}}.$$

If the thin samples are standard with a unit thickness (for example, *d*_1_ = 1 mm), then *d*_1_^1/*m*^*E_BD_*_1_ = *E*_1_. Getting rid of the subscript of *M* in Equation (40) gives:(41)$$E_{BD}=E_{1}d^{-\frac{1}{m}}.$$

Since 1/*m* > 0, *E_BD_* will decrease as *d* increases. Equation (41) is exactly the same as that derived in Equation (34). This is what is betrayed from the perspective of the Weibull distribution.

Making a logarithmic transformation for (41) and letting *C* = lg*E_BD_*_0_ give:(42)$$\mathrm{lg}E_{BD}=C_{1}-\frac{1}{m}\mathrm{lg}d.$$

Equation (42) means that lg*E_BD_* is linear to lg*d* with a slope rate of −1/*m*. Thus, the value of *m* can be known conveniently by fitting the *E_BD_* vs. *d* data linearly in a log-log coordinate system.

Figure 13 shows the fitting results of two types of PMMA samples under nanosecond pulses [70]; one is pure, the other is porous. It can be seen that the *m* of the pure PMMA is large (7.4), but the *m* of the porous PMMA is small (3.8). It is worth mentioning that an *m* of 3.8 is close to the theoretical value of m = *4* in the last section. It is also worth mentioning that the value of *m* is affected by the dielectric quality obviously. Based on the research in [70], the better the dielectric quality is, the larger *m* is.

### 5.2. Expectation and Standard Error of Weibull Distribution

The value of *m* not only reflects the dielectric quality, but also the breakdown characteristics. In Equations (41) and (42), *m* is defined as the shape parameter of the Weibull distribution. Different values of *m* mean different Weibull distributions. The breakdown probability density of the Weibull distribution in Equation (35) is:(43)$$f\left(E\right)=\frac{mE^{m-1}}{\eta }\exp\left(-\frac{E^{m}}{\eta }\right).$$

Figure 14a shows the *f*(*E*) for different values of *m*. From this figure, it can be seen that the larger the value of *m* is, the more concentrated the distribution is.

The expectation *μ* and standard error *σ* of the Weibull distribution in Equation (35) are:(44)$$\mu \left(m\right)=\eta^{\frac{1}{m}}\Gamma \left(1+\frac{1}{m}\right),$$
and
(45)$$\sigma \left(m\right)=\eta^{\frac{1}{m}}{\left[\Gamma \left(1+\frac{2}{m}\right)-\Gamma^{2}\left(1+\frac{1}{m}\right)\right]}^{\frac{1}{2}},$$
where Γ means the gamma function. The normalized *μ__N_* and *σ__N_*, which are divided by *η*^1/*m*^, are shown in Figure 14b. From this figure, it is seen that the larger *m* is, the larger *μ__N_* is(or closer *μ__N_* to 1) and the smaller *σ__N_* is (or closer *σ__N_* to 0).

### 5.3. Physical Meaning of a or 1/m

By comparing the deduced minus power relation in Equation (41) from the perspective of the breakdown mechanism and that in Equation (12) obtained by fitting the experimental result, one can easily obtain:(46)a=1/m.

Now, what is the practical meaning of *a* or 1/*m* in the minus power relation for the thickness effect of solid dielectrics? In order to answer this question, the unified standard error *σ*’ is defined, which is as follows:(47)σ’=σ/μ,

This value is similar to the standard error in the normal distribution. Figure 15a shows the comparison of *σ*’ with *a*; from the figure, it is seen that the two values are basically equal to each other, only with a deviation, *δ*, smaller than 0.03, as shown in the inset in Figure 15b. Here, *δ* is defined as (*σ*’ − *a*). This means that the minus power exponent *a* can be represented by the standard error of the *E_BD_* on a fixed thickness. In other word, *a* has the physical meaning of the standard error of *E_BD_* for a fixed thickness.

As a conclusion for this section, the minus power relation can also be deduced from the Weibull distribution. The shape parameter *m* reflects the dielectric quality. The larger *m* is, the better the quality a dielectric has. In addition, the power exponent 1/*m* or *a* has the physical meaning of the standard error of *E_BD_*.

## 6. Discussion on the Power Exponent of *a* or 1/*m* in Minus Power Relation

Here, two groups of discussion are made: one is for factors influencing *a* or 1/*m*, the other is for the information betrayed from *a* or 1/*m* once it is obtained.

### 6.1. Factors Influencing a

In Table 3, different values of *a* in the minus power relation from experiments in different conditions are summarized. Based on this table, factors such as thickness range, time scale, temperature, electrode configuration, and environmental liquid can be discussed.

#### 6.1.1. Temperature

Helgee researched the thickness effect under different temperatures [47]. By fitting each group of *E_BD_* v.s. *d* data, the value of *a* under different temperatures *T* can be obtained and plotted, as shown in Figure 16.

This figure reveals that *a* increases slightly as *T* increases in a range of *a* ≥ 0.25 (or *m* ≤ 4). It is believed that this kind of dependency can be explained from the perspective of the breakdown model of Forlani.

Based on the analysis in Section 4.2, the cathode is involved in the breakdown process when *m* ≤ 4. In addition, the electron injection mechanism is probably related to the tunnel effect when 2 < *m* ≤ 4. According to the field-induced emission formula affected by temperature:(48)$$j_{i}\left(T\right)\approx AE^{2}\exp\left(-\frac{B\theta \left(y\right)}{E}\right)\left[1+{\left(\frac{\pi k_{B}T}{D}\right)}^{2}\right],$$
where *D* is the transmittance coefficient. It is seen that the injection ability of cathode is enhanced as *T* increases, which means that the role the cathode plays in the breakdown process becomes more obvious. Based on Figure 12, if the role of cathode becomes obvious, the value of *a* should increase.

#### 6.1.2. Time Scale

When the time scale of the applied voltage increases, the heat accumulated in the dielectric become more obvious and the breakdown mechanism gradually changes from an electronic process to a thermal process [71]. If the heat is accumulated more easily, the temperature of the cathode would increase. According to the dependence of *a* on *T* in Figure 16, *a* would increase. By reviewing the relevant literatures, some proofs are found. For example, *a* is 0.125 under 10 ns by L. Zhao et al. [61] and *a* is 0.7 under dc voltage by J. H. Mason [44]. As to these two groups of experiments, the test samples are all made of PE, the sample thickness falls in a millimeter range and the samples are all immersed in oil. Thus, they can be compared together.

#### 6.1.3. Electrode Configuration and Ambient Liquid

Based on Table 3, the effects of the electrode configuration and the ambient liquid on *a* can also be discussed. From the lines of 1991/Mason on PP with a thickness of 8–76 μm, it is seen that a more uniform electric field would result in a smaller value of *a*. From the lines of 1991/Mason on PVC with a thickness of 40–500 μm, it is seen that a higher *ε_r_* of the ambient liquid may result in a lower value of *a*. These two phenomena can be explained from the perspective of the practical meaning of *a*. When the field is more uniform, the distribution of *E_BD_* on a fixed thickness would be more concentrated, since the influence causing the divergence of the standard error becomes weak. Similarly, when *ε_r_* of the ambient liquid is high, the field tends to focus on a dielectric which has a lower *ε_r_*—i.e., the test sample. Thus, the influence of the ambient liquid causing the divergence of the standard error of *E_BD_* becomes less, and *a* becomes smaller.

#### 6.1.4. Thickness Range

In Figure 5, the distribution of *a* in a wide thickness range is plotted. This figure shows that the dependency of *a* on the dielectric thickness *d* seems “random”. It is believed that two main mechanisms may be responsible for this “random distribution”. Firstly, when the thickness gets smaller, the breakdown process is affected more seriously by the cathode, such as the Al_2_O_3_ film layers embedded in metal-oxide-semiconductor ~MOS [72,73]. Thus, the value of *a* tends to increase based on Figure 12. Secondly, when the thickness gets larger, heat can easily accumulate in the dielectric and the temperature of the cathode would increase. Thus, the value of *a* also tends to increase based on Figure 16. Aside from these two effects, other factors such as the time scale, the temperature, the electrode configuration, and the ambient liquid can all affect the specific value of *a* in a fixed group of experiments. Thus, a “random distribution” on *a* takes on.

As a conclusion of this subsection, factors such as time scale, temperature, electric uniformity, ambient liquid, and thickness range can all have influence on the value of *a*. Test conditions such as short time scale, low temperature, uniform field, and high *ε_r_* of ambient liquid can all result in a small value of *a*.

### 6.2. What Does m Betray When m > 4 (or a < 0.25)?

From Table 3, it is noted that the results of *a* < 0.25 are relatively less. *a* < 0.25 is equivalent to *m* > 4. For convenience, the case of *m* > 4 is discussed.

From Figure 12, it is seen that when *m* < 4 and the dielectric thickness is lower, the mechanism of the thickness effect is mostly related to the electron injection of the cathode. When *m* ≥ 4 and the dielectric thickness is larger, the mechanism of the thickness effect is mostly related to the electron avalanche formation. As to solid insulation structures, the thickness is usually from mm to cm range. Thus, discussion of the case of *m* ≥ 4 has more practical value. Here, this case is discussed especially.

Now, a question comes: once the value of *m* for a thick insulation material or structure is obtained, what kind of information can be betrayed? This question can be answered from three perspectives.

(1) From the perspective of mechanism: From Figure 11, it is seen that the larger the value of *m* is, the higher the *E_BD_* is. In addition, from Figure 10 it is seen that a higher *E_BD_* corresponds to a higher avalanche probability *P*. This means that the formation of the avalanche is easier for a dielectric with a larger *m* than that with a smaller *m*.

(2) From the perspective of dielectric quality: from Figure 13 it is seen that a larger *m* corresponds to a better dielectric quality.

(3) From the perspective of breakdown distribution: From Figure 14a, it is seen that the larger the value of *m* is, the more concentrated the breakdown probability density is. In addition, from Figure 14b, a larger *m* corresponds to a higher expectation and a smaller standard error, which means that the *E_BD_* is also higher.

Now, let us consider that this question from the perspective of *a*. When *a* < 0.25, the smaller the value of *a* is, the more easily the avalanche forms, and the higher the *E_BD_* is, the better the dielectric quality and the more concentrated the breakdown events are. Table 4 summarizes all the information betrayed when *m* ≥ 4 or *a* ≤ 0.25.

## 7. Conclusions

Based on the review and the analysis in the whole paper, the questions put forward in the introduction can tentatively be answered:

(1) The minus power relation is preferable to characterize the thickness effect of solid dielectrics.

(2) The physical mechanism responsible for the minus power relation of the thickness effect lies in the electron injection of the cathode combined with the avalanche process in dielectrics.

(3) The application range of the minus power relation is from several Å s to several millimeters.

In addition, the following conclusions have been drawn:

(4) The practical meaning of the minus power exponent *a* is the relative standard error of the distribution (*σ**/μ*) of *E_BD_* on a fixed thickness.

(5) The value of *a* (or 1/*m*) is different in different experiments, which is affected by factors such as time scale, temperature, electric field uniformity, ambient liquid, and thickness range. The specific value of *a* or *m* needs to be tested from experiments.

(6) A smaller value of *a* corresponds to an easy avalanche formation, a higher *E_BD_*, a better dielectric quality, and a more concentrated *E_BD_* distribution.

## Figures and Tables

**Figure 1 nanomaterials-10-02473-f001:**
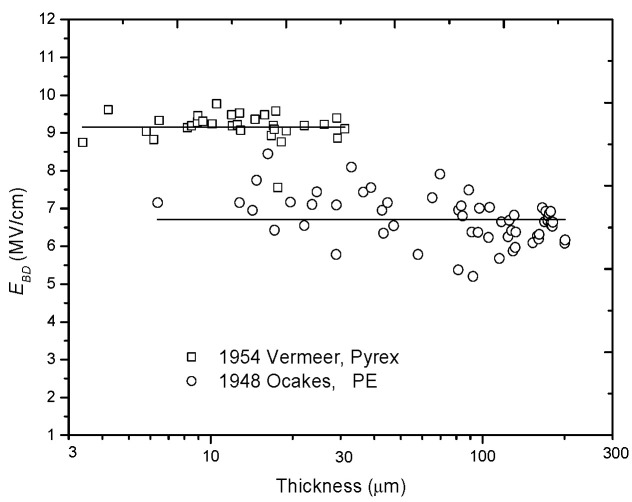
Experimental results supporting the constant relation of the dielectric field *E_BD_* with thickness *d*.

**Figure 2 nanomaterials-10-02473-f002:**
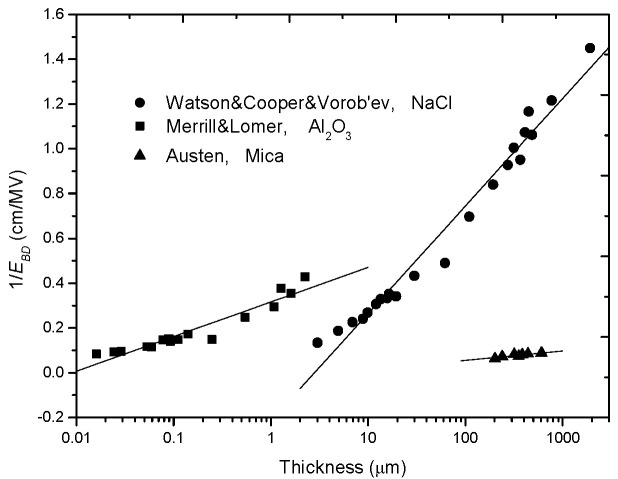
Experimental results supporting the reciprocal-single-logarithm relation of *E_BD_* with *d*.

**Figure 3 nanomaterials-10-02473-f003:**
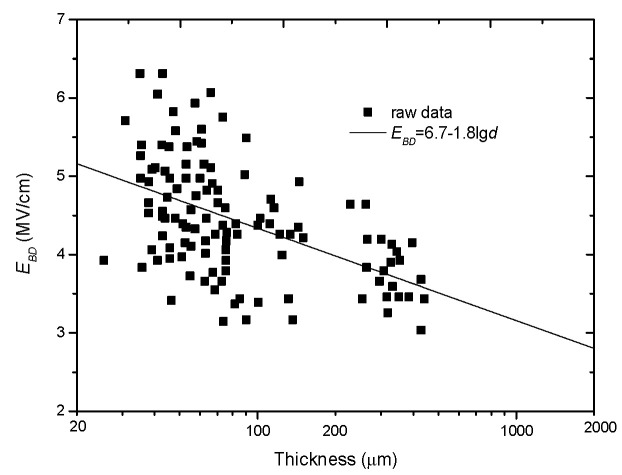
Experimental results to support the minus-single-logarithm relation of *E_BD_* with *d*.

**Figure 4 nanomaterials-10-02473-f004:**
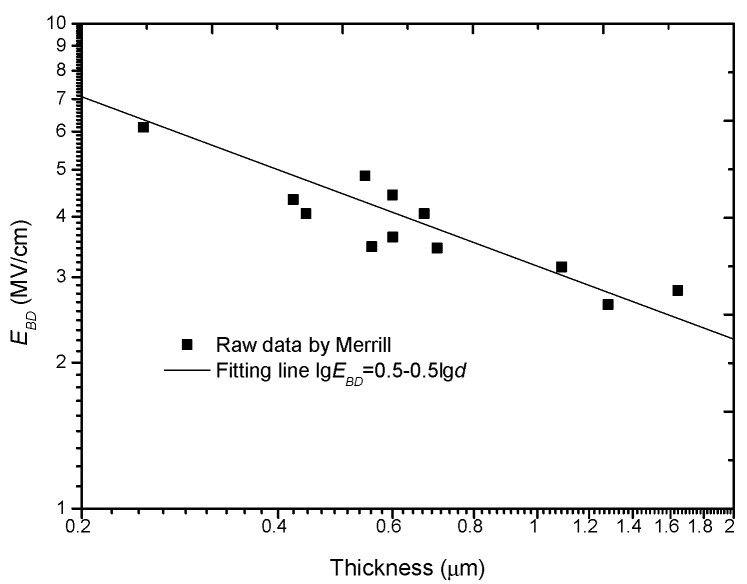
Experimental results to support the double-logarithm relation of *E_BD_* with *d*.

**Figure 5 nanomaterials-10-02473-f005:**
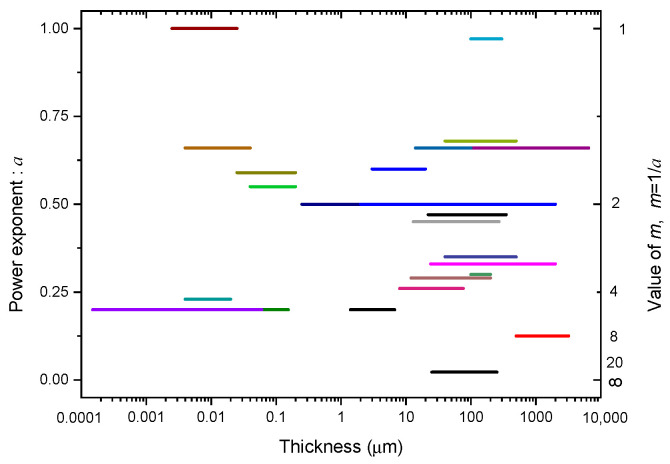
The distribution of *a* or *m*(=1/*a*) in a wide thickness range based on the data summarized in Table 3.

**Figure 6 nanomaterials-10-02473-f006:**
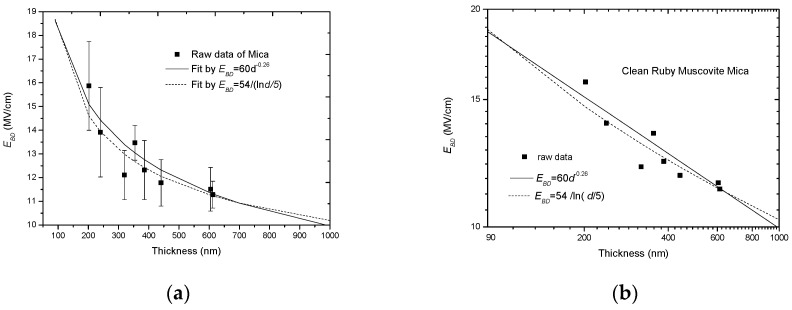
Comparison between the minus power relation and the reciprocal-single-logarithm relation with raw data from Austen [30]. (**a**) Two types of fitting in a linear coordinate system, (**b**) two types of fitting in a log-log coordinate system.

**Figure 7 nanomaterials-10-02473-f007:**
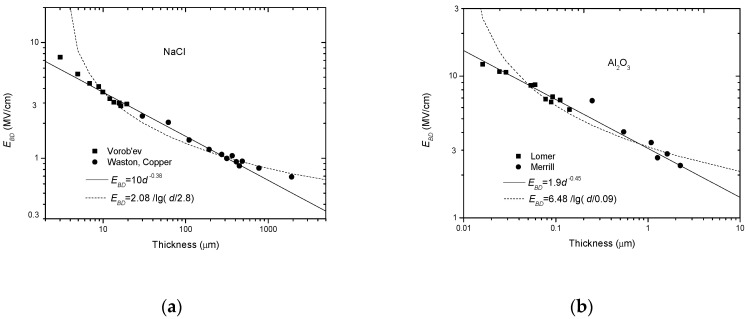
Comparison between the minus power relation and the reciprocal-single-logarithm relation with experimental data from O’Dwyer in a log-log coordinate system [33]. (**a**) NaCl, (**b**) Al_2_O_3_.

**Figure 8 nanomaterials-10-02473-f008:**
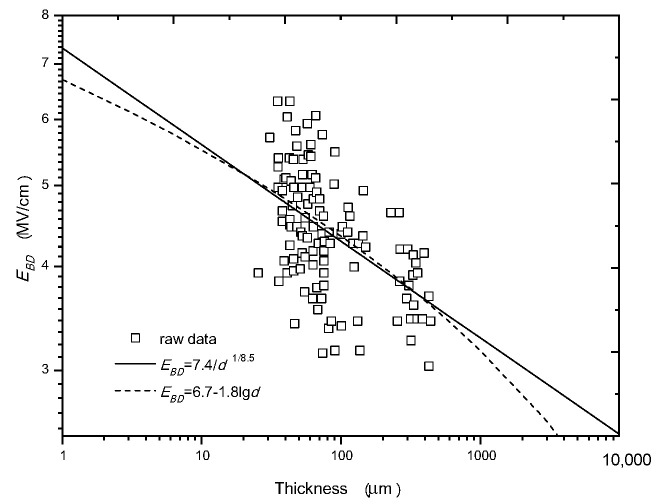
Comparison between the minus power relation and the minus-single-logarithm relation with the experimental data from Cooper in a log-log coordinate system.

**Figure 9 nanomaterials-10-02473-f009:**
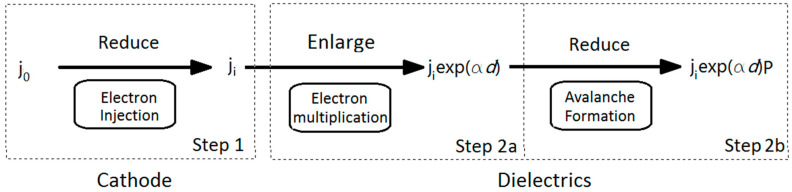
Schematics of the breakdown model by F. Forlani.

**Figure 10 nanomaterials-10-02473-f010:**
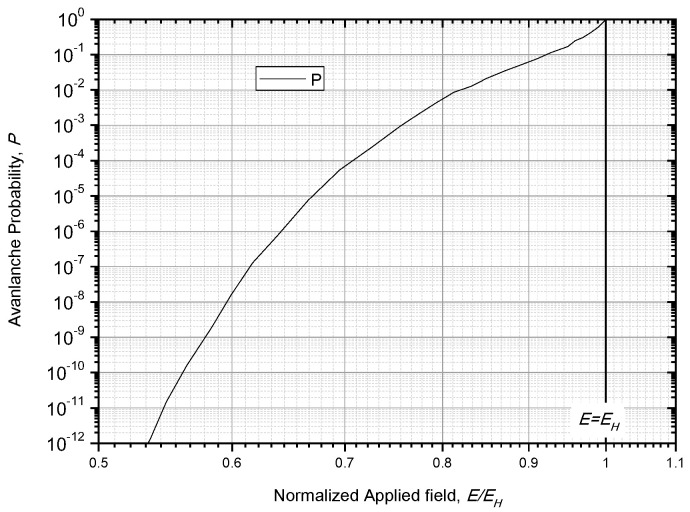
The dependency of avalanche formation probability *P* on the normalized applied field *E/E_H_* [40,41].

**Figure 11 nanomaterials-10-02473-f011:**
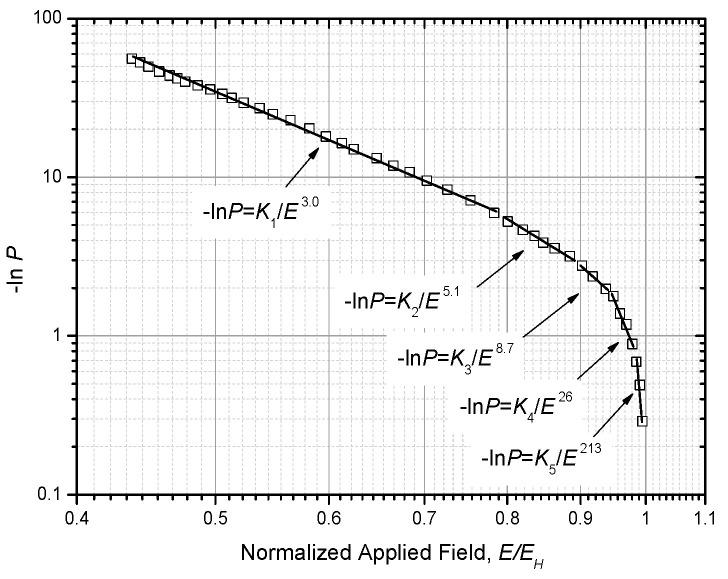
Plot of the effect of −ln*P* on *E*/*E*_H_ in a log-log coordinate system.

**Figure 12 nanomaterials-10-02473-f012:**
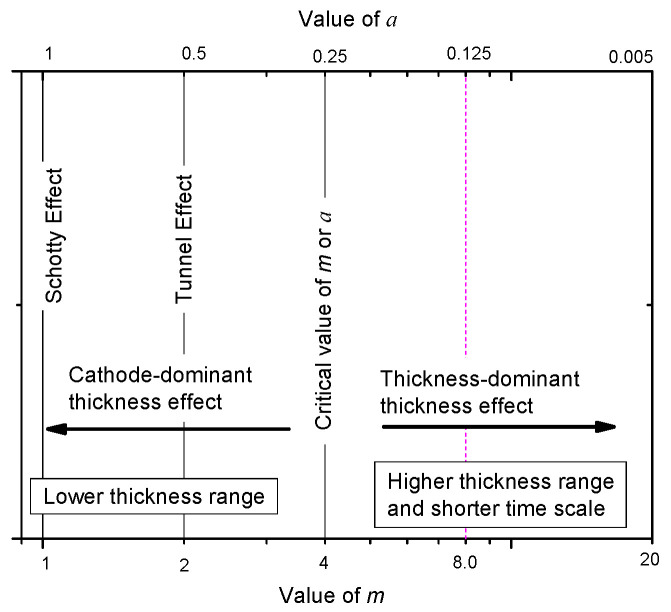
Different breakdown mechanisms for different values of *m* betrayed by the minus power relation.

**Figure 13 nanomaterials-10-02473-f013:**
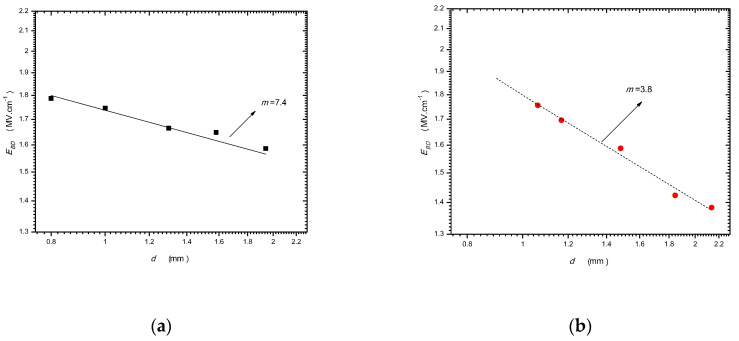
Fitting of different groups of *E_BD_* v.s. *d* data from two types of PMMA samples. (**a**) Pure PMMA; (**b**) porous PMMA.

**Figure 14 nanomaterials-10-02473-f014:**
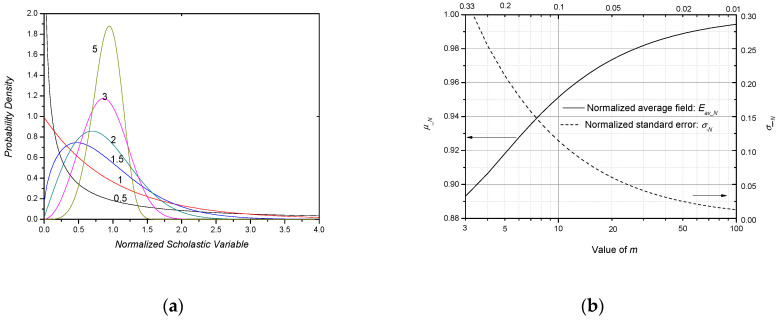
Weibull distribution for different values of *m*. (**a**) Breakdown probability density v.s. scholastic variable for different *m*; (**b**) normalized expectation and standard error of Weibull distribution dependent on *m*.

**Figure 15 nanomaterials-10-02473-f015:**
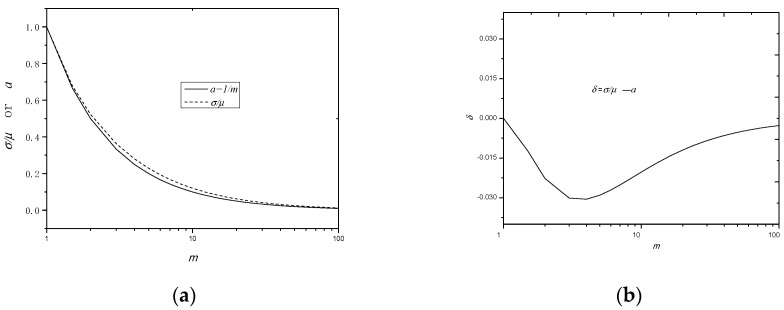
Comparison of the standard error *σ/μ* with the value of *a* (or 1/*m*). (**a**) *σ/μ* and *a* (or 1/*m*) are dependent on *m*; (**b**) *δ* (=*σ/μ* − *a*) is dependent on *m*.

**Figure 16 nanomaterials-10-02473-f016:**
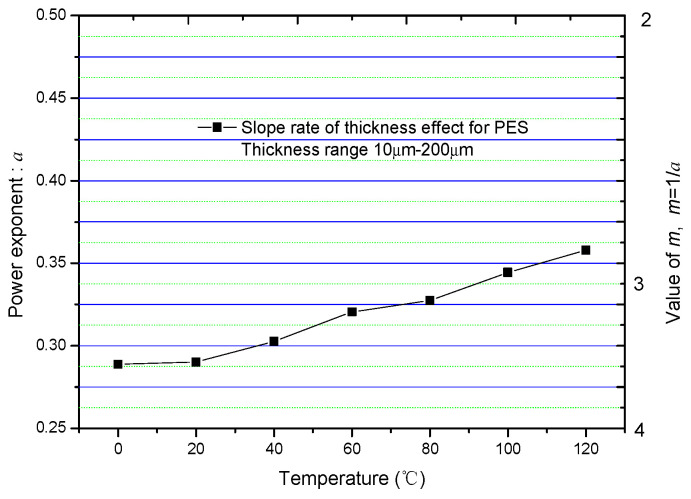
The dependence of *a* on temperature.

**Table 1 nanomaterials-10-02473-t001:** Summary of classical breakdown theory.

Breakdown Type	General Mechanism	Basic Characteristics
Intrinsic breakdown	“Electron instability”	1. *E_BD_* is independent of *d*. 2. The breakdown time is in the nanosecond time scale. 3. The breakdown happens in a low temperature range.
Avalanche breakdown	Electron impact and ionization	1. *E_BD_* is dependent on *d* and the electrode configuration. 2. The breakdown time is in the nanosecond time scale. 3. The breakdown happens in a low temperature range.
Thermal breakdown	“Heat instability”	1. The breakdown time is longer (in the microsecond time scale or longer). 2. *E_BD_* is related to sample and electrode waveform. 3. The breakdown happens in a high temperature range.
Electro-mechanical breakdown	Electro-mechanical force	1. It is common for plastics and crystals. 2. It happens easily when defects exist in dielectrics.

**Table 2 nanomaterials-10-02473-t002:** Four typical relations for the thickness effect of solid dielectrics.

Typical Relation	Mathematical Expression	Mechanism	Researcher/Year
Constant relation	$E_{BD}\left(d\right)=C$	Intrinsic breakdown	Oakes/1948 [6] Vermeer/1954 [37]
Reciprocal-single-logarithm relation	$1/E_{BD}\left(d\right)=A\mathrm{lg}d-B$	Avalanche breakdown	Austen/1940 [30] O’Dwyer/1967 [33]
Minus-single-logarithm relation	$E_{BD}\left(d\right)=D-F\mathrm{lg}d.$	/	Cooper/1963 [39]
Double-logarithm relation or minus power relation	$\mathrm{lg}E_{BD}\left(d\right)=G-a\mathrm{lg}d.$	Electron injection and avalanche	Forlani/1964 [40] Merrill/1963 [42]

**Table 3 nanomaterials-10-02473-t003:** Summary of the experimental results for the relation of *E_BD_* = *E*_1_*d*^-*a*^ for the thickness effect.

Year/Researcher	Test Object and Condition	Thickness Range	Value of *a*	Comments/Feature
1948/Oakes	PE, ac	22 μm–350 μm	*a* = 0.47 [6]	
1961/Cooper	NaCl	236 μm–544 μm	*a* = 0.33 [55]	
1963/Vorob’ev	NaCl	3 μm–20 μm	*a* = 0.60 [55]	
1965/Watson	NaCl	24 μm–2000 μm	*a* = 0.33 [55]	In mm range.
1950/Lomer	Al_2_O_3_	13 nm–0.154 μm	*a* = 0.20 [55]	
1963/Merrill	Al_2_O_3_	0.25 μm–2.5 μm	*a* = 0.50 [42]	
1968/Nicol	Al_2_O_3_	0.15 nm–60 nm	*a* = 0.20 [55]	In Å range.
1955/Mason	PE, 1/25 μs pulse	0.1 mm–6.5 mm	*a* = 0.66 [56,57]	In mm range.
1971/Agarwal	Bariμm stearate, dc	2.5 nm–25 nm	*a =* 1.0 [54]	*a* is the largest.
1971/Agarwal	Bariμm stearate, dc	25 nm–200 nm	*a =* 0.59 [54]	
1979/Yoshino	Hexatriacontane, 6 μs	14 μm–100 μm	*α =* 0.66 [51]	
1982/Singh	MgO, ac	4 nm–20 nm	*a =* 0.23 [48]	In nm range.
1983/Singh	La_2_O_3_, ac	4 nm–40 nm	*a =* 0.66 [48]	
1983/Baguji	TiO_2_, ac	40 nm–200 nm	*a =* 0.55 [49]	
1991/Mason	PP, ac, φ63.5 mm	8 μm–76 μm	*a* = 0.24 [44]	Reflecting the factor of electrode on *E_BD_*_._
1991/Mason	PP, ac, φ12.5 mm	8 μm–76 μm	*a* = 0.33 [44]	
1991/Mason	PP, ac, φ10 mm v.s. φ10 mm	100 μm–500 μm	*a* = 0.5 [44]	
1991/Mason	PVC, dc, *ε_r_* of liquid is 9.	40 μm–500 μm	*a* = 0.33, 0.38 [44]	Reflecting factor of ambient liquid.
1991/Mason	PVC, dc, *ε_r_* of liquid is 5.	40 μm–500 μm	*a* = 0.66, 0.70 [44]	
1992/Helgee	PI, ac	13 μm–27 0μm	*a* = 0.39 [58]	
1992/Helgee	PEI, ac	13 μm–270 μm	*a* = 0.44 [58]	
1992/Helgee	PET, ac	13 μm–270 μm	*a* = 0.47 [58]	
1992/Helgee	PEEK, ac	13 μm–270 μm	*a* = 0.48 [58]	
1992/Helgee	PES, ac	13 μm–270 μm	*a* = 0.51 [58]	
1996/Yilmaz	PES, ac	12 μm–200 μm	*a =* 0.26–0.32 [59]	
1997/Yilmaz	PES, ac (0 °C)	100 μm–200 μm	*a =* 0.28 [47]	Focusing on the factor of temperature.
1997/Yilmaz	PES, ac (80 °C)	100 μm–200 μm	*a =* 0.30 [47]	
1997/Yilmaz	PES, ac (120°C)	100 μm–200 μm	*a =* 0.32 [47]	
2003/Yang	TiO_2_, dc	100 μm–300 μm	*a =* 0.97 [50]	*a* is the largest.
2004/Theodosiou	PET, dc	25 μm–350 μm	*a =* 0.50 [52]	
2010/Diaham	PI, dc	1.4 μm–6.7 μm	*a* = 0.16–0.25 [60]	
2011/Zhao	PMMA, PE, Nylon, PTFE *, ns pulse	0.5 mm–3.2 mm	*a =* 0.125 [61]	In nanosecond pulse
2012/Chen	PE, dc	25 μm–250 μm	*a =* 0.022 [53]	*a* is the smallest.
2013/Neusel	Al_2_O_3_, TiO_2_, BaTiO, SrTiO_3_	2 μm–2 mm	*a =* 0.5 [62]	Plenty of dielectrics were tested.
2013/Neusel	PMMA, PS **, PVC ***, PE	2 μm–2 mm	*a =* 0.5 [62]	

* PTFE: Poly tetra fluoroethylene; ** PS: Polystyrene; *** PVC: Polystyrene.

**Table 4 nanomaterials-10-02473-t004:** Information betrayed from *m* when *m* ≥ 4 for thick dielectrics (or information betrayed from *a* when *a* ≤ 0.25).

Perspective	A Larger *m* (or a Smaller *a*) Means:
Breakdown mechanism	A larger *P* and a larger *E_BD_*.
Dielectric purity	A better dielectric purity.
Statistics	A higher *E_BD_* and a smaller *σ*

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
