# Peer review of "Review and Mechanism of the Thickness Effect of Solid Dielectrics"

_nanomaterials, 2020, doi:10.3390/nano10122473_

Round 1

Reviewer 1 Report

Electric breakdown is a physical threshold, not yet fully understood, that limits the range of applicability of dielectrics materials. The main problem is to establish a clear relation between the maximum applicable "breakdown voltage" with the dielectric thickness, and this review aims at providing an interesting perspective into this matter.

In this well compiled contribution, the Authors have described their point of view on the state of the art on dielectric breakdown and the relation with thickness, as emergeing from several experimental and theoretical treatments proposed in literature. However, since thickness is the main player of this article, I am left with a big doubt:

What thickness?

In am led to this doubt because I find the title and keywords of this article somehow in conflict. In the title the Authors talk of "solid dielectrics", leading to think that they are describing stiff materials. However in the keywords the talk of "polymers", which leads to think that they include in their review also soft materials. While for stiff materials, due to small deformations, it is legitimate to talk of only one thickness, for soft materials the situation is rather different, since for example in soft electroactive polymers the unstressed thickness and the deformed thickness can be rather different. Here elastic deformations will play a major role.

I thus suggest elucidating this matter; it is not possible to talk of "soft polymers" while excluding the fundamental role played by elastic deformations and, henceforth, of the strong variation of thickness due to electro-mechanical sources. Furthermore, for soft polymers there exist several failure mechanisms on top of "electric breakdown", and in fact it is not even clear how does electric breakdown differ from other mechanisms, like "pull-in" or "electro-mechanical instability". 

I recommend that the Authors make reference to the following two works for further details on the relevance of thermo-mechanical and elastic effects in soft polymers:

- Electro‐thermal and ‐mechanical model of thermal breakdown in multilayered dielectric elastomers, L. R. Christensen, O. Hassager, A.L. Skov, AIChE Journal, 66(8) (2020).

- Catastrophic Thinning of Dielectric Elastomers, G. Zurlo, M. Destrade, D. DeTommasi, G. Puglisi, PRL 118, 078001 (2017).

Reviewer 2 Report

The paper presents an exhaustive review of the influence of thickness of solids in electric breakdown. It is very useful, and clearly written. There are some minor corrections to be made, in particular:

  1. There is a typo in line 145.
  2. There is a typo in line 146, in units. The same in Table 3.
  3. In Table 3 , and also in some sentences a space must be between values and units.
  4. In figure 8 the dispersion of data is very high, so the conclusion is dubious. A comment about this would be useful.
  5. There is a typo in line 223.
  6. Figure 15 is very important, so must be improved. Check also d in the inset.
  7. There is a typo in line 433, α and not a. Also in the caption.
  8. Section 6.1.2, is not convincing. There is only 2 points, so I propose delete figure 17, and discussing only with these 2 points, referring this fact.
  9. There is a typo in line 471, α and not a.
  10. There is a typo in line 505.
